# Acquired cancer resistance to combination immunotherapy from transcriptional loss of class I HLA

K.G. Paulson[1,2,3], V. Voillet[2], M.S. McAfee[2], D.S. Hunter[2], F.D. Wagener[2], M. Perdicchio[2,6], W.J. Valente [2], S.J. Koelle[1,2], C.D. Church[1], N. Vandeven[1], H. Thomas[1], A.G. Colunga[1], J.G. Iyer[1], C. Yee[4], R. Kulikauskas[1], D.M. Koelle [1,2,5], R.H. Pierce[2], J.H. Bielas[1,2], P.D. Greenberg[1,2], S. Bhatia[1,2,3], R. Gottardo [1,2], P. Nghiem [1,2,3] & A.G. Chapuis[1,2,3]

Understanding mechanisms of late/acquired cancer immunotherapy resistance is critical to improve outcomes; cellular immunotherapy trials offer a means to probe complex tumor–immune interfaces through defined T cell/antigen interactions. We treated two patients with metastatic Merkel cell carcinoma with autologous Merkel cell polyomavirus specific CD8+ T cells and immune-checkpoint inhibitors. In both cases, dramatic remissions were associated with dense infiltration of activated CD8+s into the regressing tumors. However, late relapses developed at 22 and 18 months, respectively. Here we report single cell RNA sequencing identified dynamic transcriptional suppression of the specific HLA genes presenting the targeted viral epitope in the resistant tumor as a consequence of intense CD8-mediated immunologic pressure; this is distinguished from genetic HLA-loss by its reversibility with drugs. Transcriptional suppression of Class I loci may underlie resistance to other immunotherapies, including checkpoint inhibitors, and have implications for the design of improved immunotherapy treatments.

[1] University of Washington, Seattle, WA, USA. [2] Fred Hutchinson Cancer Research Center, Seattle, WA, USA. [3] Seattle Cancer Care Alliance, Seattle, WA, USA. [4] MD Anderson Cancer Center, Houston, TX, USA. [5] Benaroya Research Institute, Seattle, WA, USA. [6] Present address: Roche, Basel, Switzerland. These authors contributed equally: P. Nghiem, A.G. Chapuis. Correspondence and requests for materials should be addressed to A.G.C. (email: achapuis@fredhutch.org)

mmunotherapy has recently entered the cancer mainstream with the widespread use of immune checkpoint inhibitors (ICIs)[1–4]. However, despite many impressive responses, the majority of cancers treated are either unresponsive or develop late/acquired resistance[5–7]. Understanding resistance is critical but complex, as tumor–immune interfaces include multiple cell populations and many target antigens[8]. Among the small number of cancers for which resistance mechanisms have been conclusively determined, genetic loss of antigen presentation to CD8 + T cells has often been identified[9]. Intriguingly, a recent report suggested that, in low antigen burden tumors, genetic loss of a single human leukocyte antigen (HLA) allele is associated with checkpoint inhibitor resistance, supporting the concept that T cells recognizing very few epitopes may mediate an immunotherapy response[10]. However, most tumors resistant to checkpoint inhibitor immunotherapy lack a readily identifiable genetic means of resistance, suggesting transcriptional (and potentially reversible) escape mechanisms may be at play.

Adoptive cellular immunotherapy for solid tumors offers a defined T cell population and a defined antigen, and we thus hypothesized that detailed longitudinal investigation of patients who developed late/acquired resistance to autologous endogenous T cell therapy combined with ICIs might help broadly inform immunotherapy resistance. We focused on patients with Merkel cell carcinoma (MCC), an aggressive skin cancer typically caused by the Merkel cell polyomavirus (MCPyV)[11–13], because of the immunotherapy responsiveness[6,14,15], exceptionally low mutational/neoepitope burden[16–18] and highly expressed, defined conserved viral antigens[11,19,20]. We first interrogated tumors from a discovery/index patient: a 59-year-old man with widely metastatic heavily pre-treated MCC whom we treated with autologous ex vivo expanded CD8+ T cells recognizing a newly described HLA-B restricted allele of MCPyV followed by checkpoint inhibitors. After a 22 month response, tumors relapsed. The targeted antigen, infused T cells, and immunohistochemistry staining for pan-HLA-ABC were all present, rendering the mechanism of escape occult. We then performed single cell RNA sequencing that revealed selective loss of *HLA-B* at the time of acquired resistance, which we found to be transcriptional and reversible. In a second validation patient, treated with HLA-A restricted CD8+ T cells and ICIs, MCC relapsed after an 18 month response with transcriptional loss of *HLA-A*, supporting the reproducibility of this escape. Transcriptional suppression of Class I loci may underlie resistance to other immunotherapies, including checkpoint inhibitors, and has implications for the design of effective immunotherapy combinations to rescue patients from late/acquired immunotherapy resistance.

## Results

### Treatment of discovery patient with MCPyV-specific T cells.
Using a peptide pool based screening approach, we identified a novel HLA-B*3502-restricted epitope in MCPyV sT-Ag oncoprotein (MCPyV-sT$_{83-91}$; Methods; Supplementary Fig. 1)[20]. A 59-year-old man expressing HLA-B*3502[+] (referred for simplicity as discovery patient, 2586-4) presented with widely metastatic MCPyV-associated MCC that was refractory to >5 prior therapies, including one infusion of pembrolizumab, an anti-PD-1 ICI[6]. The patient underwent leukapheresis, from which MCPyV-sTAg-specific polyclonal endogenous CD8+ T cells were expanded by stimulation with autologous dendritic cells pulsed with the MCPyV-sT$_{83-91}$ (NCT01758458; Supplementary Fig. 2)[21]. Several metastases were irradiated to upregulate pan class I HLA (Methods; Supplementary Fig. 3)[21,22], with remaining disease unmanipulated for monitoring. MCPyV-specific

CD8+ T cells were infused ($10^{10}$ cells per m$^2$) on day 0 and again 33 days later. Toxicities included transient fevers lasting <24 h, requiring only supportive care on the general ward (Supplementary Table 1). Since his tumors enlarged, pembrolizumab and ipilimumab[23], an anti-CTLA-4 ICI, were added, followed by regression (Fig. 1a, b, Supplementary Fig. 3). Tumors shrank by >90%, with symptomatic improvement, and the patient returned to work. Twenty-two months after T cell infusion and in the context of ongoing ICI therapy, the patient relapsed, indicating late/acquired immunotherapy resistance.

**Infused CD8+ T cells persisted in peripheral blood at relapse**. During the clinical response, infused, virus-specific CD8+ cells persisted in the peripheral blood (Fig. 1c, Supplementary Fig. 4) and no other virus-specific CD8+ or CD4+ T cells recognizing alternate epitopes in MCPyV were detected (Supplementary Fig. 5)[19,20]. At relapse, antigen-experienced MCPyV-specific CD8 + cells abundantly persisted in the peripheral blood (>25% of CD8+ cells) and their immunophenotype was similar to the time of response (Fig. 1c, Supplementary Fig. 4).

**Tumor resistance mechanism remained unclear by standard evaluations**. Biopsy of a relapsed tumor (new tumor on lower leg; day + 832; not previously irradiated) revealed preserved expression of HLA-ABC and of MCPyV oncoproteins by immunohistochemistry (Fig. 1d). DNA sequencing (whole exome and MCPyV) detected several tumor-associated/somatic mutations (e.g., *PTEN*) in the pre-treatment tumor, but no additional mutations explaining immune escape at acquired resistance, including no mutations or loss of-heterozygosity in the *HLA-B* gene, sequenced *HLA-B* promoter region, or targeted MCPyV epitope (Fig. 1d, Supplementary Data 1, Supplementary Table 2). Given the absence of an identifiable genomic basis, we explored transcriptional regulation as a mechanism for tumor escape.

**scRNAseq of blood revealed T cell activation at response**. We first assessed the activity of infused T cells by performing single cell RNA sequencing (scRNAseq) with whole-transcriptome expression analysis on serial PBMCs using the 10x Genomics platform[24] ($n = 11,021$ cells; Methods). Overlay of analyses at four time-points (pre-treatment, early post treatment day + 27, responding post treatment day + 376, relapse/acquired resistance post treatment day + 614) revealed overlap of all clusters, indicating similarity between the processed cells and endogenous CD8+ T cells (Supplementary Fig. 6). Unsupervised clustering distinguished CD8+, CD4+, NK, B, and dendritic cells, and monocytes (Fig. 2a, Supplementary Fig. 7)[25]. Three CD8+ T cell clusters were identified: naïve/central memory, effector memory/effector, and an activated effector population significantly enriched at response, which overexpressed glycolysis (*GAPDH*, mitochondrial RNAs) and other activation (*IL32*; actin) transcripts relative to the effector memory/effector cells (Fig. 2b–d; Supplementary Table 3)[26–28], while maintaining an expression profile otherwise consistent with traditional effector CD8+ T cells (expression of granzymes and perforins without *CCR7* or *IL7R* expression; Supplementary Fig. 7).

**Activated CD8+ T cells infiltrated the responding tumor**. In tumor biopsies at time of treatment response and concurrent to the presence of activated CD8+ cells in blood (day + 349), CD8+ T cells expressing the activation marker HLA-DR newly infiltrated the previously T cell "cold", now shrinking MCC (Fig. 3a, b)[29,30]. The dominant CD8+ cell clonotype in the infusate identified by its unique CDR3-beta was also dominant in regressing metastases (Fig. 3c)[31], implicating infused CD8+ cells

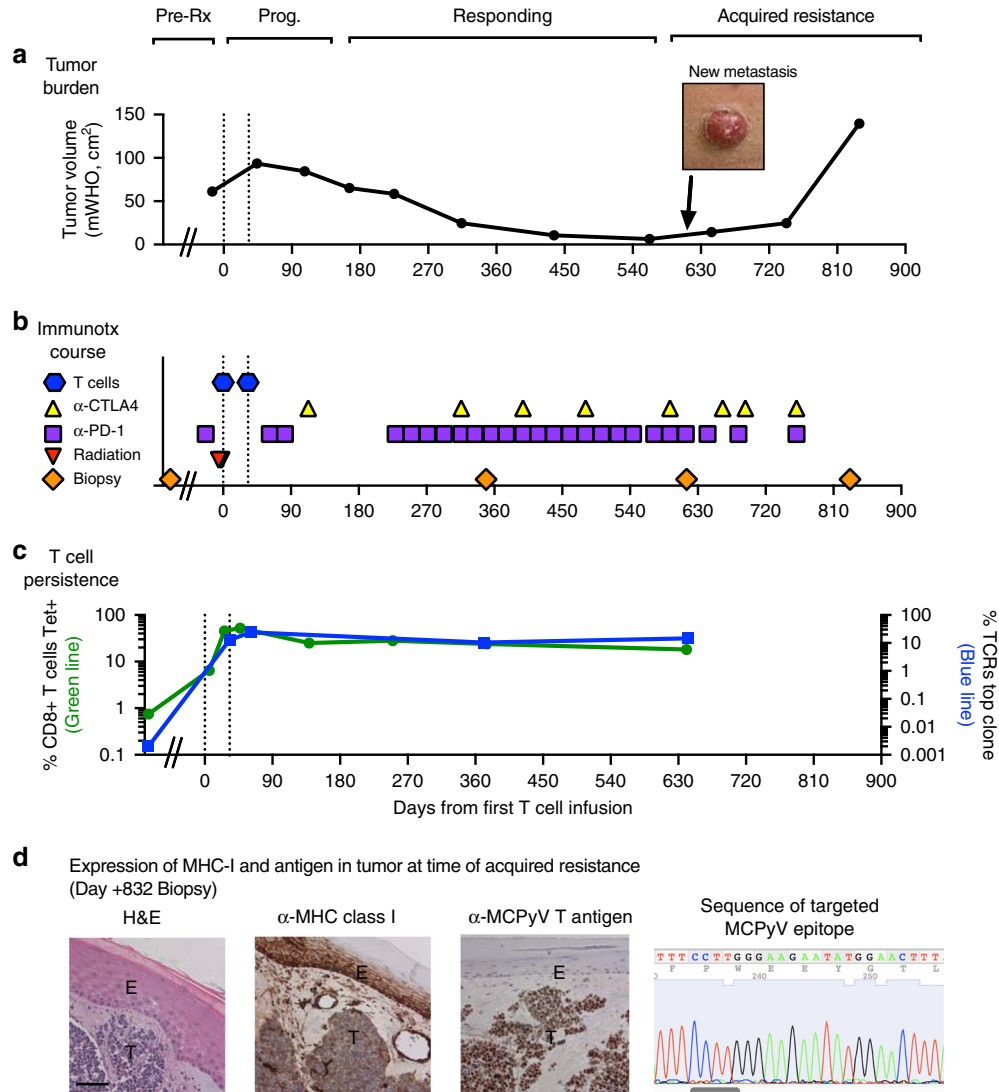

**Fig. 1** Acquired resistance to combination immunotherapy. T cell infusions indicated by dashed lines. **a** Clinical course. **b** Immunotherapy treatments. **c** T cell persistence. CD8+ T cell persistence by MCPyV-sT$_{83-91}$ tetramer is on left axis (green line) and frequency of dominant infused clone by CDR3-beta sequencing on right axis (blue line; this clonotype was 85% of the infusion product). **d** Evaluation of tumor at acquired resistance/late relapse (post-treatment day + 832): Left: Immunohistochemistry demonstrating expression of MHC class I (pan-HLA–ABC) in tumor (T) and epidermis (E) and MCPyV T antigen in tumor. Scale bar indicates approximately 100 micrometers. Right: sequence of the targeted MCPyV-sT$_{83-91}$ epitope was unmutated. Discovery patient (2586-4) is shown

as the likely mediators of tumor regression. At a later biopsy after acquired resistance (day + 832), CD8+ T cells were absent from the relapsing tumor. This finding, combined with the presence of quiescent T cells in the peripheral blood identified by scRNAseq, suggests tumor specific CD8+ T cells were no longer encountering antigen (Figs. 2 and 3).

**scRNAseq of relapsed tumor revealed *HLA-B* downregulation**. To define the mechanism of late/acquired resistance, scRNAseq was performed on viably frozen tumor digests (Fig. 4a; *n* = 7431 cells), from pre-treatment and acquired resistance (D + 615) time points. Although bulk/unsorted cells were assessed together, scRNAseq tSNE unsupervised clustering permitted segregation and simultaneous analysis of tumor, macrophage, B lymphocyte, T lymphocyte, fibroblast and endothelial populations (Supplementary Fig. 8). The transcriptome of non-tumor clusters before and at late/acquired resistance were superimposable, implying changes in these microenvironment cells were not driving

immune evasion. In contrast, MCC tumor cells demonstrated marked transcriptional change, visualized by distinct spatial separation of pre-treatment and relapsed tumor cells in tSNE plots (Fig. 4a, b), with 255 differentially expressed genes (Methods; Supplementary Data 2). These included many expected to be overexpressed in a multiply-recurrent cancer (e.g., cell cycle), but also defined the immune escape. Specifically, MCC tumor significantly downregulated *HLA-B*, but not *HLA-A* at acquired resistance (Fig. 4c, d, Supplementary Fig. 9). This implies intense immunologic and selective pressure from the transferred HLA-B*3502-restricted CD8+ cells. *HLA-B* loss was exclusive to tumor cells and undetectable by standard HLA-ABC immunohistochemistry (Fig. 1d). To exclude sampling bias, tumor from a second post-resistance biopsy (day + 832) was obtained and qPCR reaffirmed *HLA-B* downregulation (Fig. 4e).

**HLA-B transcription was restored with hypomethylating agents**. We next sought to determine whether transcriptional

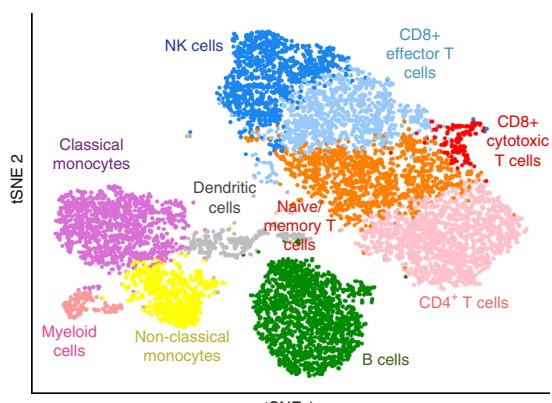

**a** Peripheral blood mononuclear cells (n = 11,021)

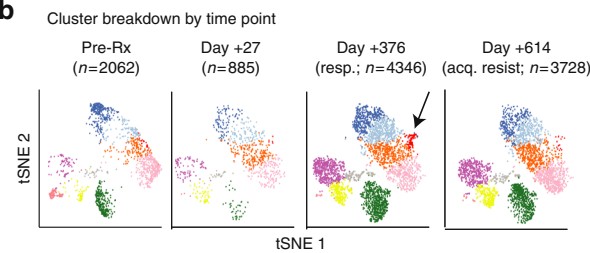

**b** Cluster breakdown by time point

Pre-Rx (n = 2062)   Day +27 (n = 885)   Day +376 (resp.; n = 4346)   Day +614 (acq. resist; n = 3728)

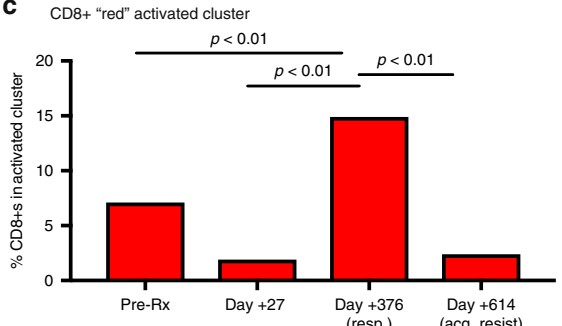

**c** CD8+ "red" activated cluster

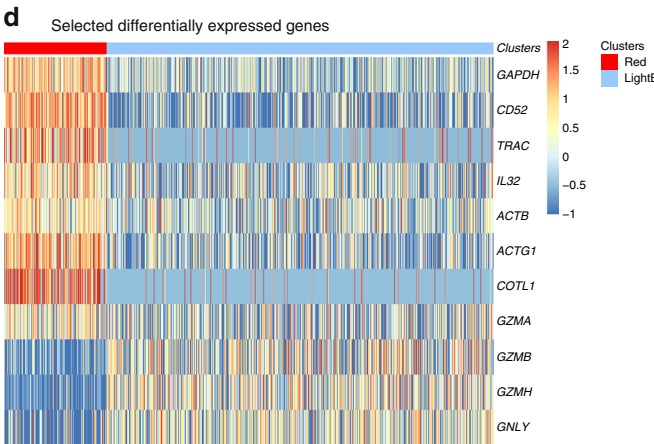

**d** Selected differentially expressed genes

**Fig. 2** scRNAseq of PBMC identifies an activated CD8+ T cell population at response. Four peripheral blood time points are shown, all from the discovery patient (2586-4): pre-treatment, early post-treatment (day + 27), treatment response (day + 376), and late/acquired resistance (day + 614). **a** t-Stochastic Neighbor Embedding (tSNE) visualization of clustering of peripheral blood. Peripheral blood mononuclear cells (PBMC; n = 11,021) clustered into populations as indicated. Representative marker genes shown in Supplementary Fig. 7. **b**, **c** Enrichment of cluster of activated CD8 + lymphocytes (red cluster, arrow) at response to immunotherapy. Clustering biostatistical analysis described in detail in the Methods, and proportion of CD8+s in each cluster at the various time points compared with Fisher's exact test. **d** Heat map of selected significantly differentially expressed genes comparing CD8+ T cells in the red activated cluster (n = 170) to those in the blue effector/EM cluster (n = 429) at response (day + 376). For full list of all 45 differentially expressed genes, see Supplementary Table 3

**Identification of and clinical course of second validation patient**. To validate our findings, we sought to perform similar analyses on a second patient, preferably treated with MCPyV-specific T cells restricted to a different class I HLA. Out of all patients previously treated at our institution with both MCPyV-specific T cells and ICIs, we identified one additional patient with early response and late/acquired immunotherapy resistance. Patient 9245-3, referred to as validation patient for simplicity, was treated with MCPyV-specific CD8+ T cells and checkpoint inhibition, this time with T cells restricted to HLA-A2 and avelumab (anti-PD-L1) as checkpoint inhibition (Supplementary Fig. 10; NCT0258482)[14,15]. This validation patient had an 18 month response to combined immunotherapy, during which he achieved complete regression by radiographic and pathologic assessment. However, at the 18 month time point, he relapsed at a single tumor site, and tumors were excised (day + 565 post-treatment). Resistant tumor expressed MCPyV T antigen and had detectable pan-HLA-ABC by immunohistochemistry (Supplementary Fig. 10).

**T cells infiltrated tumor during treatment response in the validation patient**. We interrogated the T cell infiltration in the microenvironment for the validation patient. Prior to T cell therapy, the tumor was T cell cold (Supplementary Fig. 11). However, during response (day + 14), the tumor became T cell hot with intratumoral enrichment of tumor-specific CD8+ T cells (Supplementary Fig. 11). At acquired resistance (day + 565), CD8 + T cells were no longer infiltrating intratumorally but instead were excluded to the tumor periphery. At that time, infused HLA-A2 restricted T cell persisted at low levels in the microenvironment (Supplementary Fig. 11).

**scRNAseq of tumor revealed *HLA-A* transcriptional down-regulation**. For the validation patient, tumor tissue was only available in viably frozen format from a time of late/acquired resistance (day + 565). Therefore, scRNAseq using the 10x genomics 5′ platform was performed on this resistant tumor (n = 5397 cells), along with matched peripheral blood (n = 5870 cells; day + 559; Fig. 4f-h, Supplementary Fig. 12) to provide additional comparative tissue. We again observed selective transcriptional downregulation of the targeted HLA (*HLA-A*; Fig. 4g) in the tumor cells, without changes in the *HLA-A* DNA sequence as determined by whole exome sequencing (Supplementary Table 4). Attempts to culture tumor from the validation patient were unsuccessful in both short term and long-term cultures, and thus reversibility with interferon-γ and 5-azacitidine could not be tested.

downregulation of HLA-B was reversible. We were able to successfully establish short term ex vivo cultures but not a cell line. In ex vivo cultures, *HLA-B* loss was reversible with either pharmacologic doses of interferon-γ or the hypomethylating agent 5-azacitadine, consistent with transcriptional downregulation[22,32].

## Discussion

Late/acquired resistance stands as a barrier to immunotherapy cure. We performed detailed investigation on two patients who had received T cell immunotherapy along with ICIs; these patients both had sustained immunotherapy responses followed by late/acquired resistance. We observed infiltration of infused CD8+ T cells into shrinking MCC tumors, supporting T cell mediated regression. In both cases, when tumors relapsed, there was apparent selective transcriptional downregulation of the HLA restricting the targeted MCPyV epitope.

Immune avoidance by genetic loss of single or all class I HLAs has been described as a mechanism of resistance to cellular immune therapies[33] and anti-PD-1 checkpoint inhibitors[9,10].

Immunotherapy escape by genetic HLA loss is important to distinguish from immunotherapy escape by transcriptional HLA loss as we observed here. In the former, the HLA alleles are deleted and new T cell responses must necessarily be targeted to alternate HLAs to overcome immunotherapy resistance. In the latter, tumor-specific HLA suppression is potentially reversible with drug therapy. Transcriptional suppression of all class I HLA genes in a coordinated fashion has been described previously by our group and others for MCC[22,32]. This has also been described as a mechanism of melanoma early[34] and, in a single case, late immunotherapy resistance[35]. Differential transcriptional suppression of the targeted class I HLA genes as a mechanism of late immunotherapy resistance demonstrates immunotherapy responses can be driven by T cells restricted to a single HLA. Additionally, such resistance cannot be readily detected by pan-HLA-ABC immunohistochemistry, indicating this mechanism might have been underappreciated previously.

Our study had limitations. Attempts to generate tumor-derived cell lines for additional functional studies were unsuccessful on both patients. We could not differentiate if acquired resistance represented immunoediting, i.e., outgrowth of a pre-existing previously rare or quiescent clone, or was new transcriptional suppression that developed after immunotherapy. In either scenario, immune pressure from the transferred CD8+ T cells revealed selective HLA downregulation that was transcriptional, and in at least one patient reversible.

Here we employ scRNAseq on thousands of cells to demonstrate strong immune pressure mediated by ICIs and transferred CD8+ T cells recognizing a single tumor epitope, and identify a novel mechanism of immunotherapy escape by selective transcriptional loss of the targeted HLA under T cell pressure, which is rendered possible based on established differential regulation of HLA-A and -B genes[36]. This has therapeutic implications for rescue as it is potentially reversible (e.g., hypomethylating agents). Transcriptional loss of antigen presentation deserves further evaluation in immunotherapy resistance, as these targetable escape mechanisms are likely active in additional tumor types.

## Methods

**Identification of HLA-B*3502-restricted epitope of MCPyV.** Tumor infiltrating lymphocytes (TIL) from a testicular metastasis of a 64-year-old man with advanced MCC (separate from featured case) were non-specifically expanded for 2 weeks with IL-2 and IL-15 cytokine support and screened against peptide pools (13 mers overlapping by 9) tiling across the entirety of expressed portion of MCPyV T antigens by Elispot[20]. Confirmation of reactivity to the small T antigen pool (containing peptides spanning the C-terminal domain of MCPyV small T antigen), identification of minimal epitope as MCPyV-sT$_{83-91}$ (Supplementary Fig. 1), and restriction to HLA-B*3502 was determined by interferon-gamma intracellular cytokine stain, using autologous irradiated PBMC pulsed with peptide at final concentration of 1 microgram per milliliter as antigen presenting cells. This peptide was then confirmed to also be held by HLA-B*3501 and HLA-B*3503 alleles, which are highly similar to HLA-B*3502. For sequence comparison analyses, all Merkel cell polyomavirus small T antigen protein sequences available in NCBI were downloaded on August 21, 2017 in FASTA format and aligned using MUSCLE[37].

**Clinical protocols.** All clinical investigations were performed in compliance with the Declaration of Helsinki principles. The first treated patient (primary focus of manuscript, 2586-4) was enrolled on protocols #2586/NCT01758458 (T cell infusion treatment), #6585 and #1765 (biological sample collection) at the Fred Hutchinson Cancer Research Center (FHCRC, Seattle, WA), all research activities were approved by the FHCRC Institutional Review Board and the Food and Drug Administration. The patient provided written informed consent. The second, "validation" patient was enrolled on protocol #9245/NCT0258482 as well as the sample collection protocols as above (9245-3). The third patient from whom the HLA-B*3502-restricted epitope was identified was enrolled on protocol #6585 (biological sample collection) and also provided written informed consent.

The primary patient (2586-4) received hypofractionated radiation for HLA upregulation to some but not all disease sites: 21 Gy in 6 fractions to pelvic nodes, 8 Gy in 1 fraction to retroperitoneum (previously heavily irradiated and recurrent post radiation), 8 Gy in 1 fraction to one of the thigh tumors. Other sites of disease, including orbit, mediastinum (bulky), bladder, lumbar spine, and other lower extremity tumors were not irradiated. Twenty-four hours thereafter the first of his two infusions of HLA-B3502 restricted CD8+ T cells targeting Merkel cell polyomavirus was administered. The second infusion was 33 days after the first. Cell dose was at 10 billion cells per m$^2$ for a total dose of 26 billion cells. After initial progression, pembrolizumab followed by ipilimumab were added. Please see Fig. 1 for schematic of infusions. Tumor volumes were determined by mWHO criteria[38].

The second validation patient (9245-3) is a 59-year-old man with metastatic MCC that had initially presented as stage IIIB disease, now metastatic at multiple sites. He had developed multiple relapses that had previously been treated with radiation; trial interventions were first systemic therapy. He received avelumab (anti-PD-L1) 10 mg/kg every two weeks[15], and four infusions of MCPyV-specific T cells at dosages ranging from 0.8-3.9 billion (dose target of 10$^{10}$ cells per m$^2$ (23.5 billion) was not met for technical reasons). Two HLA-upregulation interventions were performed as detailed in Supplementary Fig. 10 including injection of intralesional interferon-beta 0.1 mg once into a single tumor lesion prior to first cycle of infusions, and irradiation of a single pelvic lymph node with an 8 Gy radiation fraction prior to second infusion. Cells were of two specificities: HLA-B35/FPW, and HLA-A0201 recognizing the "KLL" epitope[39], however only HLA-A0201 cells persisted.

**Generation of tetramer.** Allophycocyanin-conjugated MCPyV-specific antigen pMHC multimers (FHCRC Immune Montoring Core Facility) were used to confirm purity of the cellular product as well as detect transferred CTL in PBMCs, with a staining sensitivity of 0.05% of total CD8+ T cells[19].

**Isolation and expansion of MCC-specific CTLs.** For the primary patient described (2586-4), peripheral blood mononuclear cells (PBMCs) were collected by leukapheresis, and all ensuing ex vivo manipulations were performed in the clinical Good Manufacturing Practices Cell Processing Facility at the FHCRC. Patient PBMCs were collected by leukapheresis and depleted of CD25+ T cells to eliminate regulatory T cells[31]. Cells were cultured with cytokines (IL-2, IL-7, IL-15 and IL-21) stimulated twice for 7–10 day cycles with autologous DCs pulsed with the HLA B*3502-restricted Merkel cell polyomavirus (MCPyV) MCPyV-sT$_{83-91}$ epitope peptide FPWEEYGTL[21,23]. Cultures containing >5% specific CD8+ cells (as assessed by tetramer binding) were GMP flow-sorted and then expanded to sufficient numbers for infusion using two cycles of the Rapid Expansion Protocol[21,23,40]. Cell products bound the MCPyV sT-Ag epitope MCPyV-sT$_{83-91}$ peptide-HLA tetramer and secreted IFN-gamma when exposed to the cognate antigen (Supplementary Fig. 2).

For the validation patient (9245-3), identical methods were used, with the exception that a second culture was also performed using the MCPyV-T antigen (sT$_{15-23}$ and LT$_{15-23}$) peptide KLLEIAPNC as stimulus. This is restricted to HLA A*0201, as previously described[39]. Cells of both specificities were mixed together immediately prior to infusion in a 1:1 ratio.

**Flow cytometry on patient peripheral blood samples.** Blood samples were collected at the indicated time points, PBMCs isolated through the research cell bank at FHCRC by standard Ficoll-Hypaque gradient, and viably cryopreserved. Cells were analyzed by flow cytometry after permeabilization, fixation and staining with fluorochrome-conjugated monoclonal antibodies to CD4 (SK3; Becton Dickinson), CD8 (3B5; Invitrogen), CD19 (H1B19; Becton Dickinson), CD16 (3G8; Becton Dickinson), CCR7 (G043H7; Biolegend), CD45RO (UCHL1; Becton Dickinson), CD28 (CD28.2; Biolegend), KLRG1 (SA231A2; Biolegend), CD27 (L128; Biolegend), CXCR3 (G025H7; Biolegend), CD127 (A019D5; Biolegend), PD1 (ED12.2H7; Biolegend), Lag3 (2DS223H; eBioscience), 4-1BB (4B4-1; Biolegend), BCL2 (100; Biolegend), CTLA-4 (L3D10) and the above described tetramers. Cells were analyzed on a Fortessa cytometer (Becton Dickinson) and data analysis performed with FlowJo. Staining, acquisition and analyses were performed on all samples in a batch on same day and negative controls included.

**Epitope spreading assessments.** For the analyses of epitope spreading in Supplementary Fig. 5, PBMCs were cultured for 72 h with MCPyV peptide pools (13

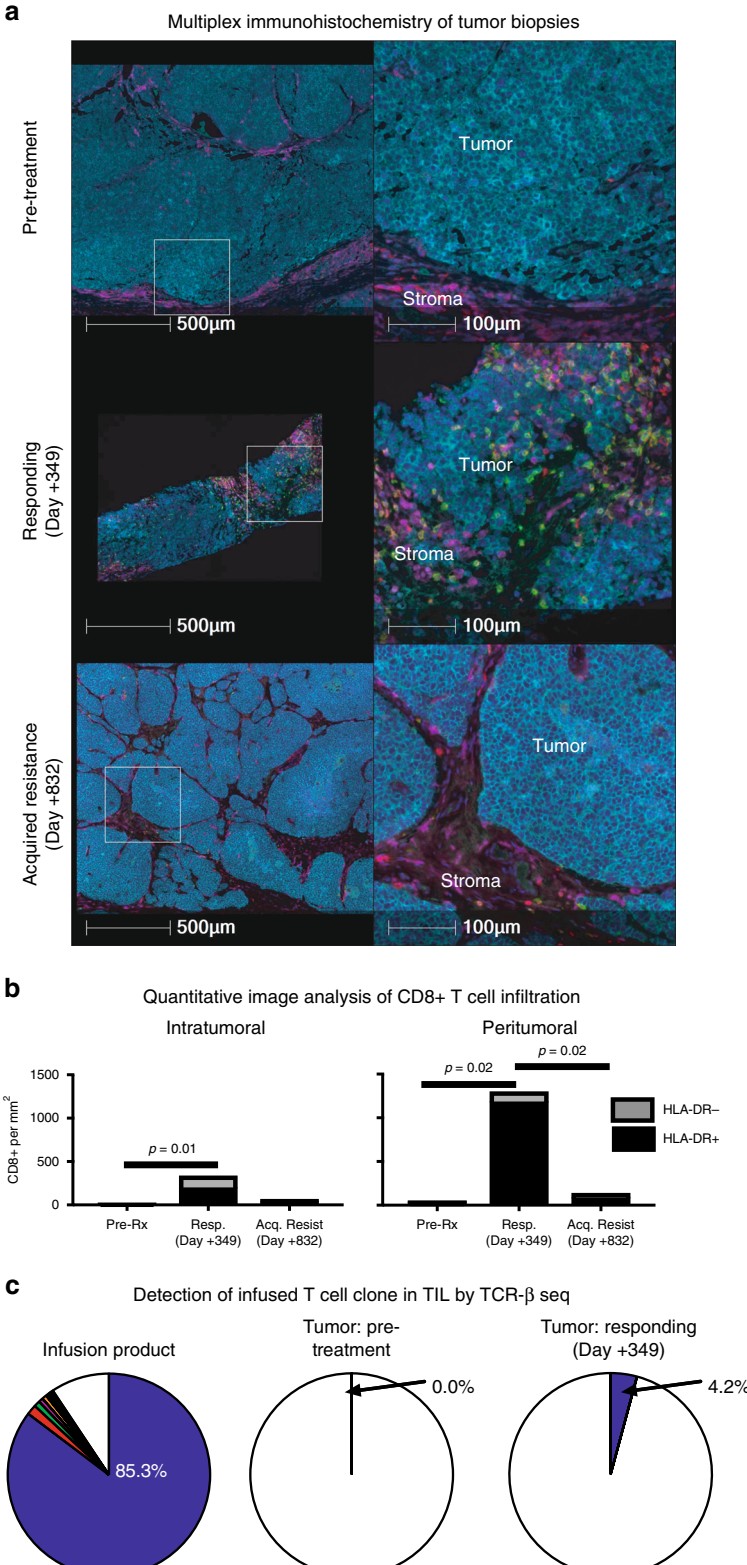

**Fig. 3** Activated CD8+ T cell infiltration into tumor at the time of treatment response. **a** Multiplex immunohistochemistry showing representative peritumoral and intratumoral CD8+ infiltrates. Arrows indicate CD3+CD8+ cells (not all indicated). Dark blue = DAPI (nuclei), Light blue = CD56 (MCC tumor), Red = CD3+CD8−, Green = CD8+, Purple = HLA-DR. Three timepoints are shown from the discovery patient (2586-4): pre-treatment, post-treatment response (day + 349), and late/acquired resistance (day + 832). **b** Quantification of density of peritumoral and intratumoral CD8 cells. Three representative microscope scan areas were digitally scored for each patient, and significant differences in density determined with student's T test. **c** Detection of infused T cell clonotypes in tumor at time of treatment response by sequencing of TCR CDR3-beta

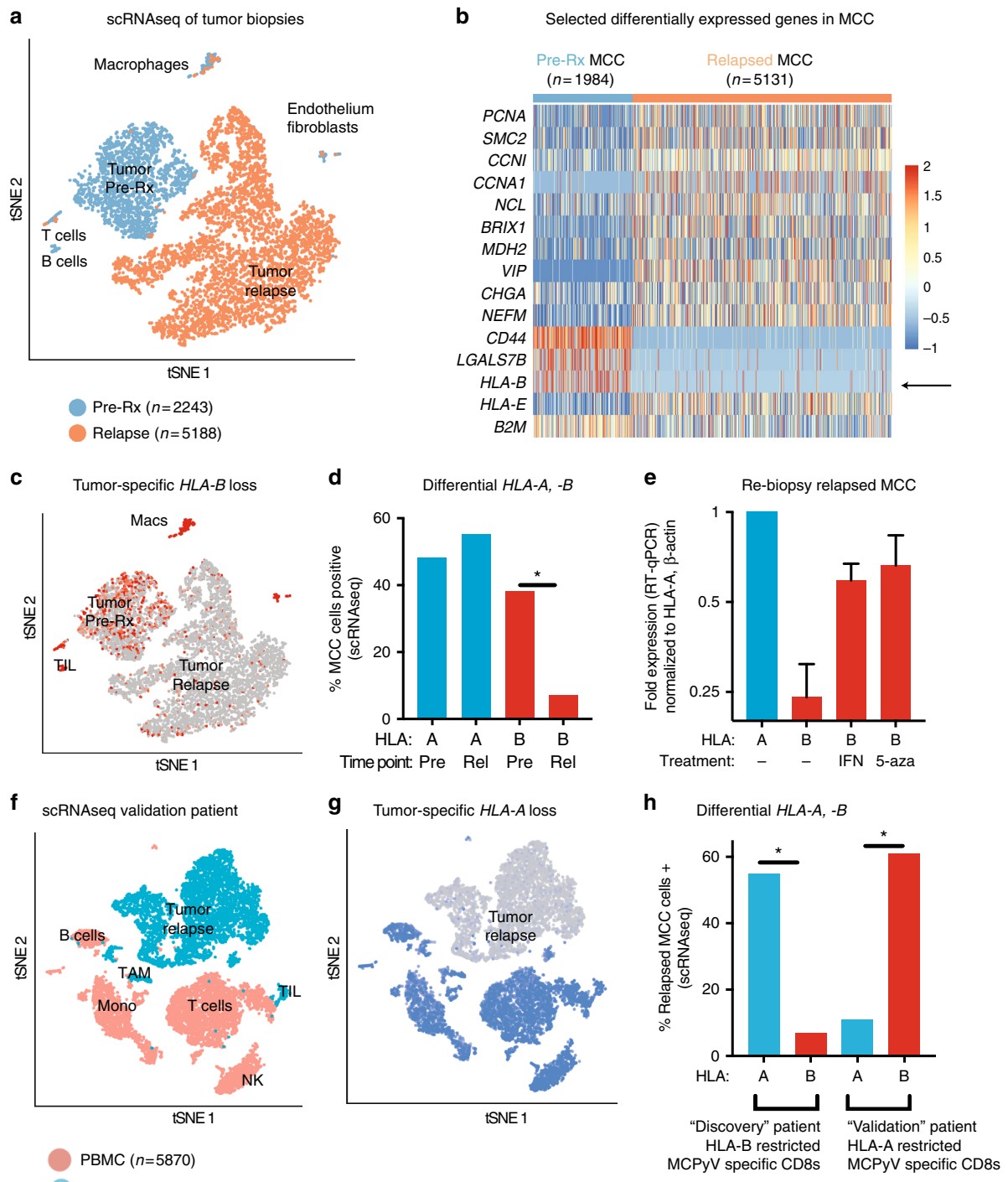

**Fig. 4** scRNAseq of tumor biopsies. **a–e** Discovery patient (2586-4). **f, g** Validation patient (9245-3). **h** Both patients. **a** tSNE of viably frozen cells (n = 7431) from tumor biopsies obtained pre-treatment (blue) and at late relapse/acquired resistance (day + 615; orange). Marked spatial separation of tumor cells from the two timepoints indicates substantial transcriptional change. **b** Heat map of selected significantly differentially expressed genes (DEGs) in tumor clusters. For full table of DEGs, see Supplementary Data 2. **c** tSNE of HLA-B expression. **d** Differential change in scRNAseq expression of *HLA-A, -B* by tumor cells. *HLA (HLA-A, HLA-B)* and time point (pre: pre-treatment, rel: acquired resistance/relapse) are indicated below. **e** qPCR validation and reversibility of *HLA-B* downregulation on a repeat tumor biopsy at time of acquired resistance, day + 832, graphed on a log2 scale. Tumor cells were cultured ex vivo for 48 h before RNA collection with either vehicle control (–), interferon gamma (1000 IU/mL; IFN) or the hypomethylating agent 5-azacytidine (1 μM; 5-aza) as indicated below. Bars represent mean and error bars represent range of two experimental runs, each performed with triplicate wells. **f** tSNE of specimens from late relapse in the validation patient (day + 565 post treatment). Viably frozen cells (n = 11267) from PBMC (pink) and tumor biopsy (blue) at late relapse/acquired resistance. Patient had received HLA-A-restricted CD8+ T cells with 18 months response followed by relapse (Supplemental Figures 10 and 11). **g** tSNE of HLA-A expression. **h** Differential expression of targeted and non-targeted HLA on relapsed tumor for both patients. Proportion MCC cells expressing gene by scRNAseq are indicated. Data for discovery patient (2586-4) are reproduced from panel 4d for clarity. Please see Methods section for details of scRNAseq biostatistical analysis and determination of DEGs

mers overlapping by 9)[20], and responses detected with interferon gamma intracellular cytokine stain.

**TCR beta CDR3 sequencing.** DNA was extracted from tetramer-sorted infusion product and unsorted PBMCs using a Qiagen DNEasy Blood and Tissue kit per manufacturers instruction. These samples and tumor biopsies were submitted to Adaptive Biotechnologies (Seattle, WA) for DNA amplification and sequencing of TCRB CDR3 using the immunoSEQ platform[41,42].

**Immunohistochemistry.** Formalin-fixed paraffin-embedded tissues were sectioned and baked per standard protocol.

MCPyV large T antigen was detected using the CM2B4 antibody (Santa Cruz, dilution 1 :50) and HLA-ABC using clone EMR8-5 (MBL, dilution 1 :1500 for slides from discovery patient and dilution 1 :8000 for slides from validation patient; antibody lots titered on normal tissues with endothelial cells serving as on-slide positive control)[22,43]. For multiplex immunohistochemistry, slides were dewaxed and stained on a Leica Bond Rx stainer using Leica Dewax solution, antigen retrieval, and antibody stripping and rinsing after each step (bond wash solution). A high stringency wash was performed after the secondary and tertiary applications using high-salt TBST solution (0.05 M Tris, 0.3 M NaCl, and 0.1% Tween-20, pH 7.2–7.6). OPAL Polymer HRP Mouse plus Rabbit (PerkinElmer, Hopkington, MA) was used for all secondary applications. Antigen retrieval and antibody stripping steps were performed at 100 °C with all other steps at ambient temperature. Endogenous peroxidase was blocked with 3% $H_2O_2$ for 8 min followed by protein blocking with TCT buffer (0.05 M Tris, 0.15 M NaCl, 0.25% Casein, 0.1% Tween 20, pH 7.6 +/- 0.1) for 30 min. The first primary antibody (position 1) was applied for 60 min followed by the secondary antibody application for 10 min and the application of the tertiary amplification reagent (PerkinElmer OPAL fluor) for 10 min. The primary and secondary antibodies were stripped with retrieval solution for 20 min before repeating the process with the second primary antibody (position 2) starting with a new application of 3% $H_2O_2$. Antibodies included CD8 (clone 144B; Dako; opal fluor 520, concentration 0.2 microgram/mL), CD56 (clone 123c3.d5; Bio SB; opal fluor 540, 1 ug/mL), CD3 (clone SP7; Thermo, opal fluor 650, concentration 1 :400), and HLA-DR (clone EP96; Bio SB; opal fluor 690; concentration 0.125 ug/mL). Slides were removed from the stainer and stained with Spectral DAPI (Perkin Elmer) for 5 min, rinsed for 5 min, and coverslipped with Prolong Gold Antifade reagent (Invitrogen/Life Technologies, Grand Island, NY). Slides were cured for 24 h at room temperature, then representative images from each slide were acquired on PerkinElmer Vectra 3.0 Automated Imaging System. Images were spectrally unmixed using PerkinElmer inForm software and exported as multi-image TIFF's for analysis. Quantitative image analysis was performed in HALO software (Indica Labs, Corrales, NM). For each slide, three representative sections were scored. CD56 staining readily identified tumor. Layers were manually drawn for intratumoral (within tumor borders) and peritumoral (edge of tumor to 100 micron beyond tumor border) regions. Three sections were counted for each slide and comparisons between time points were made using the Student's *t* test.

**Exome sequencing.** For both patients 2586-4 (discovery) and 9245-3 (validation), DNA was isolated from PBMC as germline control, and from tumor time point pre-immunotherapy and at acquired resistance as tumor specimens (DNA from frozen tumor specimens), using a Qiagen QIAAmp DNA extraction kit under manufacturer's recommended conditions. Quality was confirmed with spectrophotometry and Qubit (Fisher Biosciences). For patient 2586-4 samples were submitted for exome sequencing through the FHCRC Genomics Core. Library prep was performed with standard procedures using Agilent SureSelect Human All Exon V6 and sequencing performed on an illumina HiSeq 2500 per Agilent recommendations. Samples were aligned to hg19 reference using annovar[44] and somatic mutations determined using MuTect[45]. For patient 9245-3, samples were submitted for exome sequencing through the University of Washington Northwest Clinical Genomics Laboratory research sequencing service. Library preparation was with xGen technology and sequencing on an illumina HiSeq 4000 at 100x coverage. Samples were aligned to hg19 with BWA[46] and variants identified with GATK[47] as per the NCGL standard pipeline.

**Generation of single cell tumor digests.** Tumor material not required for clinical pathology analysis was placed in RPMI. Tumor was mechanically minced with scissors and forceps into small pieces and then resuspended in 20 mL freshly prepared digestion medium (20 mL RPMI plus 0.002 g DNAse (Worthington Biochemical) plus 0.008 g collagenase (Worthington Biochemical) plus 0.002 g hyaluronidase (Worthington Biochemical) in a 10 cm dish. Digesting tumor was incubated in 37 degree tissue culture incubator for 3 h with occasional manual rocking. After incubation, tumor digest was strained through a 70 micron cell strainer, cells were centrifuged, counted, and assessed for single cellularity and viability, and resuspended in freeze medium (50% human serum, 45% RPMI, 5% DMSO). Cells were frozen overnight at -80C in a styrofoam sandwich then transferred to liquid nitrogen for long-term storage.

**Single cell RNA sequencing (scRNAseq).** For samples from patient 2586-4, cells were thawed, washed, and labeled in a single cell fashion using the 10x genomics 3'

Chromium v2.0 platform[24] as per manufacturer's instructions. Library preparation was performed as per manufacturer's protocol with no modifications. Library quality was confirmed with illumina TapeStation high sensitivity (evaluates library size), qubit (evaluates dsDNA quantity), and KAPA qPCR analysis (KAPA Biosystems, evaluates quantity of amplifiable transcript). Samples were mixed in equimolar fashion and sequenced on an illumina hiSeq 2500 "rapid run" mode according to standard 10x genomics protocol.

For samples from patient 9245-3, scRNAseq was performed using the 10x Genomics 5' V(D)J and gene expression chromium platform, with cell washing, barcoding, and library preparation as per manufactures instruction. Library quality was confirmed as above. Samples were sequenced on an Illumina NovaSeq 6000 (gene expression) and HiSeq 4000 (V(D)J) as per 10x genomics protocol for this instrument.

**Transcriptome alignment, barcode assignment and UMI counting.** The Cell Ranger Single-Cell Software Suite (versions 2.0.0 and 2.1.0 for the discovery and validation patients respectively) were used to perform sample demultiplexing, barcode processing and single-cell gene counting (http://10xgenomics.com/). First, raw base BCL files were demultiplexed using the Cell Ranger *mkfastq* pipeline into sample-specific FASTQ files. Second, these FASTQ files were processed individually using the Cell Ranger *count* pipeline, which made use of the STAR software[48] to align cDNA reads to the hg38 human reference genome (Ensembl) and the Merkel cell polyomavirus sequence (HM011556.1). Aligned reads were then filtered for valid cell barcodes and unique molecular identifiers (UMIs). Cell barcodes with 1-Hamming-distance from a list of known barcodes were considered. UMIs with sequencing quality score >10% and not homopolymers were retained as valid UMIs. A UMI with 1-Hamming-distance from another UMI with more reads, for a same gene and a same cell was corrected to this UMI with more reads. Tumor and PBMC samples were, respectively, aggregated together using the Cell Ranger *aggr* pipeline resulting in two gene-barcode count matrices (tumor and PBMC) to be used for downstream analyses. A correction for sequencing depth was also performed during the aggregation[19].

**Biostatistical analysis—data normalization and correction.** UMI normalization was performed as in Zheng et al.[19]. Only genes with at least one UMI count detected in at least one cell were retained for analysis. A library-size normalization was performed for each cell. UMI counts were scaled by the total number of UMI in each cell and multiplied by the median of the total UMI counts across cells. The data were then log2-transformed and corrected for unwanted sources of variation (number of detected UMIs) using the *ScaleData* R function as described in the Seurat manual[49]. The corrected-normalized gene-barcode matrix was used as input for dimension reduction and clustering analysis, whereas the normalized gene-cell barcode matrix was used for the MAST analysis as described below.

**Gene expression analysis: discovery patient tumor.** Following sequence alignment and filtering, a total of 7431 tumor cells (2243 cells before and 5188 cells after T cell therapy) were analyzed. The corrected-normalized gene-barcode matrix was used to perform principal component analysis (PCA) and t-distributed stochastic neighbor embedding (tSNE) analyses. First, the top 873 most variable genes selected by Seurat (log-mean expression values greater than 0.0125 and dispersion (variance/mean) greater than 0.5) were kept for PCA. The first top 10 principal components (PCs) were then down selected for tSNE visualization. One thousand iterations of tSNE using a perplexity value of 30 were performed. Cell classification and clustering were done according to the expression of established MCC tumor markers: *NCAM1*, *ENO2*, *CHGA* and *KRT20* and also TILs markers. There were a total of 1984 cancer cells before, and 5131 cancer cells at the acquired resistance time point. Differential expression analysis between tumor cells before and after T cell therapy was performed using the R package MAST[50]. The normalized gene-cell barcode was used as input. The model included the cellular detection rate (CDR) as a covariate to correct for biological and technical nuisance factors that can affect the number of genes detected in a cell (e.g., cell size and amplification bias). Genes were declared significantly differentially expressed at a false discovery rate (FDR) of 5% and a fold-change >1.3.

**Gene expression analysis: discovery patient PBMC.** Following sequence alignment and filtering, a total of 12,874 cells were analyzed. As for the tumor samples, the corrected-normalized gene-barcode matrix was used to run PCA and t-distributed stochastic neighbor embedding (tSNE) analyses. First, the top 1203 most variable genes selected by Seurat (log-mean expression values greater than 0.0125 and dispersion (variance/mean) greater than 0.5) were kept for PCA. Again, the first top 10 PCs were then down selected for tSNE visualization. One thousand iterations of tSNE using a perplexity value of 30 were performed. Cell clustering was performed using a graph-based clustering method implemented in Seurat (*FindClusters* R function—share nearest neighbor (SNN) modularity optimization based clustering algorithm). Thirteen distinct clusters of cells were identified using the top 10 PCs with a neighborhood size of 40 and resolution of 0.6. Based on the clustering results, we removed cells belonging to three different distinct clusters enriched for expression of red blood cell and megakaryocyte markers that were likely the result of blood contamination (leaving total of 11,021 cells for analysis).

Clusters were labeled according to enrichment of specific markers (*FindMarkers* R function implemented in Seurat). This analysis resulted in nine distinct clusters: CD4+ T cells, CD8+ T cells, CD8+ effector T cells, B cells, NK cells, CD14+ monocytes, CD16+ monocytes, myeloid cells and dendritic cells. Differential expression analysis between CD8+ and CD8+ effector T cells for the third time point was performed using the R package MAST[50] as described in the previous section. R code is attached in Supplementary Data 3. Data submitted to NCBI Gene Expression Omnibus (GEO), accession GSE 117988.

**Gene expression analysis: validation patient**. The aggregated corrected-normalized gene-barcode matrix was used to first run PCA and t-distributed stochastic neighbor embedding (tSNE) analyses. The top 2450 most variable genes selected by Seurat (log-mean expression values greater than 0.05 and dispersion (variance/mean) greater than 0.5) were kept for PCA. The top 10 PCs were used for tSNE visualization. One thousand iterations of tSNE using a perplexity value of 50 were performed. As described in the previous section, cell clustering was performed using a graph-based clustering method implemented in Seurat (*FindClusters* R function). Nineteen distinct clusters of cells were identified using the top 10 PCs with a neighborhood size of 20 and resolution of 0.6. Clusters were labeled according to enrichment of specific markers. R code is attached in Supplementary Data 4, and data submitted to NCBI GEO, accession 118056.

**Treatment with hypomethylating agents**. Ex vivo tumor was mechanically dissociated, filtered and cultured for 48 h in RPMI supplemented with 20% FBS and pen/strep antibiotics. Untreated tumor was compared to tumor treated with 5-azacytidine (Toronto Research Chemicals, concentration 1 μM final) or gamma-interferon (final concentration 1000 IU/mL). RNA was isolated using directzol and reverse transcription performed. qPCR was performed using RT2 SYBR mastermix and commercially available primer sets to the indicated genes (Qiagen). Each condition was run in triplicate and qPCR repeated twice.

## Data availability

Single cell RNA sequencing data from the discovery patient submitted to National Center for Biotechnology Information Gene Expression Omnibus (NCBI GEO), accession GSE 117988 [https://www.ncbi.nlm.nih.gov/geo/query/acc.cgi?acc=GSE117988] and R code to generate tSNE plot using Seurat software package[51,52] attached in Supplementary Data 3. Single cell RNA sequencing data from the validation patient submitted to NCBI GEO, accession GSE 118056 [https://www.ncbi.nlm.nih.gov/geo/query/acc.cgi?acc=GSE118056] and R code attached in Supplementary Data 4.

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

## Acknowledgements

We thank the patients and their families, Kristina Lachance, Kieu-Thu Bui, Natalie Miller, Olga Afanasiev, Erika Kiem, Marcus Lindberg and FHCRC core facilities: Immunohistochemistry, Flow Cytometry, Cell Processing, Immune Monitoring, and Genomics.

## Author Contributions

K.G.P. contributed to the conception, design, and implementation of immunohistochemistry, flow cytometry, exome, and scRNAseq experiments, and assembly of data. K.G.P. and A.G. Chapuis wrote the manuscript. V.V., S.J.K. and R.G. conducted the biostatistical analyses. M.P. and F.D.W. performed flow cytometry. D.S.H., W.J.V., and J.H.B. performed, assisted, and provided critical infrastructure and guidance for scRNAseq, respectively. C.D.C., N.V., H.T., A.G. Colunga, M.S.M., and R.K. acquired/processed the clinical specimens and assisted in implementation of clinical trial. J.G.I. and D.M.K. identified the MCPyV-sT83-91 epitope. C.Y. developed the protocol for cellular product preparation. R.H.P. was the pathologist. P.D.G. contributed immunologic expertise and critical analysis. S.B. contributed to the protocol development, patient treatment, and collection of clinical samples. P.N. and A.G. Chapuis conceived the protocol and project, designed experiments, and served as principal investigators on the trial. Funding: NIH-5R01CA176841 and NIH-K24CA139052 (P.N.), NIH-5K08CA169485 (A.G. Chapuis), NIH-5T32CA009515 (K.G.P.), Immunotherapy Integrated Research Center at FHCRC (A.G. Chapuis), Damon Runyon (A.G. Chapuis), MCC gift fund at UW (P.N.), P30 CA015704 (FHCRC), EMD Serono, and 10X Genomics.

## Additional information

**Competing Interests:** A.G. Chapuis has received support from Juno therapeutics. P.D.G. has received support and has ownership interest in Juno Therapeutics, Immune Design, and Innate Pharma. P.N. has received consulting fees from EMD Serono, Pfizer, Merck Sharpe and Dohne, Amgen, Incyte, Takeda, Mallinckrodt and research support from EMD Serono and Bristol-Meyers Squibb. A.G.C., M.S.M., D.M.K., K.G.P., C.D.C., and P.N. and their institutions have intellectual property related to T cell receptors recognizing Merkel cell polyomavirus. S.B. has received advisory board honoraria from Genentech and EMD-Serono; his institution (University of Washington) has received research funding from EMD-Serono, Merck, BMS, Oncosec, and Immune Design. A.G. Chapuis and K.G.P. have received reagents from 10X genomics. R.G. received consulting fees from Juno Therapeutics. The authors declare no other competing interests.

