## [Peer Review File · Nature Communications]

REVIEWERS' COMMENTS:

Reviewer #1 (Remarks to the Author):

It is my opinion that the authors have fully addressed the comments from the previous reviewers. In addition they have strengthened the manuscript considerably by adding another patient case where this mechanism of immune evasion also occurred, further validating their findings.

Reviewer #2 (Remarks to the Author):

In this revised manuscript, the authors report on the detailed analysis of two Merkel cell carcinoma cases that recurred during combination immunotherapy. By leveraging unique, high-quality clinical biospecimens and cutting-edge assays (e.g. scRNAseq) they were able to find two examples where resistance to HLA-restricted autologous anti-tumor T cells was associated with selective transcriptional downregulation of the relevant HLA gene in resistant tumor cells. This mechanism of immune evasion is distinct from genetic loss of MCH class I expression and generalized HLA transcriptional downregulation that has been previously reported. The work is elegant and compelling.

Specific comments:

1. The conclusions from scRNAseq analysis of PBMCs are focused on a cluster of CD8+ T cells that the authors describe as “activated”. Can more detail be given on how these cells correlate with more traditional definitions of activated T Effector cells? If the 5' sequencing kit allow the authors to identify the cells expressing the TCR of the tetramer-reactive CD8+ cells, please indicate if they are among these “activated” cells. Is exclusion from this group how the authors determined the CD8+ T cells at the time of tumor relapse were “quiescent in the peripheral blood”?
2. In the response to reviewers, the authors clearly point out how their current findings are distinct from the genetic downregulation of HLA and from the generalized transcriptional downregulation of class I HLA. The manuscript does not effectively contrast the current findings with how MCPyV+ Merkel cell carcinoma is known to show generalized HLA downregulation that is reversible with radiation and drug treatment. In addition to the prior published works on this topic from Nghiem et al., it would be appropriate to include *Sci Rep.* 2017 May 23;7(1):2290 in the discussion.
3. “In either scenario, immune pressure from the transferred CD8 T cells revealed selective HLA downregulation that was transcriptional and reversible.” As reversibility was only demonstrated in one case, the authors should be careful not to imply that it was shown in both cases.
4. Can the authors please be explicit in the text and figure legends about the tumor biopsy timepoints that were analyzed.
5. “Affirming” in the first paragraph does not seem to be the correct use of this word. Did the authors intend to say confirming?
6. There are a few abbreviations that are not defined in the text and figure legends. E.g. IHC in the text, and the T cell subtypes in Fig S4.

Point-By-Point Response to Reviewer Comments

Reviewer #1 (Remarks to the Author):

“It is my opinion that the authors have fully addressed the comments from the previous reviewers. In addition they have strengthened the manuscript considerably by adding another patient case where this mechanism of immune evasion also occurred, further validating their findings.”

We thank the reviewer for their comments and kind review.

Reviewer #2 (Remarks to the Author):

“In this revised manuscript, the authors report on the detailed analysis of two Merkel cell carcinoma cases that recurred during combination immunotherapy. By leveraging unique, high-quality clinical biospecimens and cutting-edge assays (e.g. scRNAseq) they were able to find two examples where resistance to HLA-restricted autologous anti-tumor T cells was associated with selective transcriptional downregulation of the relevant HLA gene in resistant tumor cells. This mechanism of immune evasion is distinct from genetic loss of MCH class I expression and generalized HLA transcriptional downregulation that has been previously reported. The work is elegant and compelling.”

Specific comments:

1. *“The conclusions from scRNAseq analysis of PBMCs are focused on a cluster of CD8+ T cells that the authors describe as “activated”. Can more detail be given on how these cells correlate with more traditional definitions of activated T Effector cells? If the 5’ sequencing kit allow the authors to identify the cells expressing the TCR of the tetramer-reactive CD8+ cells, please indicate if they are among these “activated” cells. Is exclusion from this group how the authors determined the CD8+ T cells at the time of tumor relapse were “quiescent in the peripheral blood”?”*

We thank the reviewer for their insightful questions. We have added more detail in the text, to explain that these CD8+ T cells do indeed also express traditional T effector genes. We updated sentences in the Results section under the subheading “scRNAseq on blood reveals T cell activation at response”.

Page 7, line 11 now reads: “Three CD8⁺ T cell clusters were identified: naïve/central memory, effector memory/effector, and an activated effector population significantly enriched at response, which overexpressed glycolysis (*GAPDH*, mitochondrial RNAs) and other activation (*IL-32*; actin) transcripts relative to the effector memory/effector cells (Fig. 2b, Fig. 2c, Fig. 2d; Supplementary Table 3), while maintaining an expression profile otherwise consistent with traditional effector CD8+ T cells (expression of granzymes and perforins without *CCR7* or *IL7R* expression; Supplementary Fig. 7).

The peripheral blood analyses from the discovery patient were done on the 3’ Chromium sequencing kit and thus TCR information is not available. Exclusion from this group is how we determined they were quiescent in the peripheral blood.

2. *“In the response to reviewers, the authors clearly point out how their current findings are distinct from the genetic downregulation of HLA and from the generalized transcriptional downregulation of class I HLA. The manuscript does not effectively contrast the current findings*

with how MCPyV+ Merkel cell carcinoma is known to show generalized HLA downregulation that is reversible with radiation and drug treatment. In addition to the prior published works on this topic from Nghiem et al., it would be appropriate to include Sci Rep. 2017 May 23;7(1):2290 in the discussion.”

We thank the reviewer for this comment and agree that this distinction is critical. In response, we have expanded our discussion to include a full paragraph comparing and contrasting our observations/mechanism to those previously observed of genetic downregulation of HLA and generalized transcriptional downregulation of HLA. We further have added the reference from Dr. Juergen Becker's group, as specifically requested.

Page 10, line 9 now reads: “Immune avoidance by genetic loss of single or all class I HLAs has been described as a mechanism of resistance to cellular immune therapies³³ and anti-PD-1 checkpoint inhibitors.^{9, 10} Immunotherapy escape by genetic HLA loss is important to distinguish from immunotherapy escape by transcriptional HLA loss as we observed here. In the former, the HLA alleles are deleted and new T cell responses must necessarily be targeted to alternate HLAs to overcome immunotherapy resistance. In the latter, tumor-specific HLA suppression is potentially reversible with drug therapy. Transcriptional suppression of all class I HLA genes in a coordinated fashion has been described previously by our group and others for Merkel cell carcinoma.^{22, 32} This has also been described as a mechanism of melanoma early³⁴ and, in a single case, late immunotherapy resistance.³⁵ Differential transcriptional suppression of the targeted class I HLA genes as a mechanism of late immunotherapy resistance demonstrates immunotherapy responses can be driven by T cells restricted to a single HLA. Additionally, such resistance cannot be readily detected by pan-HLA-ABC immunohistochemistry, indicating this mechanism might have been underappreciated previously.”

3. *““In either scenario, immune pressure from the transferred CD8 T cells revealed selective HLA downregulation that was transcriptional and reversible.” As reversibility was only demonstrated in one case, the authors should be careful not to imply that it was shown in both cases.”*

We agree with the reviewer. We have updated the sentence in question in the discussion. Page 10, line 26 now reads: “In either scenario, immune pressure from the transferred CD8 T cells revealed selective HLA downregulation that was transcriptional, and in at least one patient reversible.”

We have also added an additional sentence to the Results section, under the heading “scRNAseq of tumor revealed HLA-A transcriptional downregulation,” to provide further clarification that we were unable to test reversibility in the second case.

Page 9, line 18 now reads: “Attempts to culture tumor from the validation patient were unsuccessful in both short term and long-term cultures, and thus reversibility with 5-azacitidine could not be tested.”

4. *“Can the authors please be explicit in the text and figure legends about the tumor biopsy timepoints that were analyzed.”*

We apologize for this omission. Certain time points were inadvertently previously only listed on figure panels and not within the text itself. We now have updated the manuscript text to explicitly identify the time points analyzed exactly as requested. For example, in the Results section

under the heading entitled “scRNAseq of blood revealed T cell activation at response,” we previously stated “overlay of analyses at four time-points revealed...”.

Page 7, line 6 now states: “Overlay of analyses at four time-points (pre-treatment, early post treatment day +27, responding post treatment day +376, relapse/acquired resistance post treatment day +614) revealed...”. Similar updates have been made throughout the Results section.

Figure legends 1, 2, 3, and 4 have all been updated to include explicit post-treatment days within the legend text, as well as the legends for supplementary figures 4, 5, 6, 11, and 12.

5. *“Affirming” in the first paragraph does not seem to be the correct use of this word. Did the authors intend to say confirming?*

We agree that we intended “confirming”. We have now removed the word affirming from the updated abstract, which has also been reworded to conform to length and journal style requirements.

6. *“There are a few abbreviations that are not defined in the text and figure legends. E.g. IHC in the text, and the T cell subtypes in Fig S4.”*

We thank reviewer 2 for finding these undefined abbreviations and have now updated the text as requested.